# Reconciling qualitative, abstract, and scalable modeling of biological networks

Loïc Paulevé [1,2 ✉], Juri Kolčák[3], Thomas Chatain [3] & Stefan Haar [3]

Predicting biological systems' behaviors requires taking into account many molecular and genetic elements for which limited information is available past a global knowledge of their pairwise interactions. Logical modeling, notably with Boolean Networks (BNs), is a well-established approach that enables reasoning on the qualitative dynamics of networks. Several dynamical interpretations of BNs have been proposed. The synchronous and (fully) asynchronous ones are the most prominent, where the value of either all or only one component can change at each step. Here we prove that, besides being costly to analyze, these usual interpretations can preclude the prediction of certain behaviors observed in quantitative systems. We introduce an execution paradigm, the Most Permissive Boolean Networks (MPBNs), which offers the formal guarantee not to miss any behavior achievable by a quantitative model following the same logic. Moreover, MPBNs significantly reduce the complexity of dynamical analysis, enabling to model genome-scale networks.

[1] Université Bordeaux, Bordeaux INP, CNRS, LaBRI, UMR5800, 351 cours de la Libération, Talence 33400, France. [2] LRI UMR8623, Université Paris-Sud, CNRS, Université Paris-Saclay, Bat 650 Ada Lovelace, Rue Raimond Castaing, Gif-sur-Yvette 91190, France. [3] Inria and LSV, CNRS (UMR 8643) and ENS Paris-Saclay, Université Paris-Saclay, 4 avenue des Sciences, Gif-sur-Yvette 91190, France. ✉email: loic.pauleve@labri.fr

Models in systems biology typically integrate knowledge and hypotheses on molecular interactions, manually or semi-automatically, gathered from experimental data found in databases and the literature. These models are often qualified as mechanistic, in opposition to those solely based on biophysical laws.

Since their introduction in the late 1960s[1,2], logical models, such as Boolean Networks (BNs), have been widely adopted for reasoning about signaling and gene networks[3–11] as they require few parameters and can easily integrate information from omics datasets and genetic screens. These models represent processes with a high degree of generalization and can offer coarse-grained but robust predictions. That makes them particularly suitable for large biological networks, for which ample global knowledge exists about potential interactions with little precise data on actual molecules abundances and reaction kinetics.

The validation of computational models is necessary to trust their subsequent predictions. In systems biology, validation primarily involves in silico reproduction of observed behaviors by executing the computational model. Such observations may be measurements of the activity, over time, or at steady state, of some of the interacting molecules under different experimental conditions. Therefore, if no executions of a BN reproduce an experimentally observed behavior (e.g., the activation of a particular gene), the model, and the associated interactions, is considered as invalid. This procedure also enables general studies on interaction motifs that are necessary or sufficient for achieving fundamental behaviors such as cellular differentiation or homeostasis[12–15].

A BN specifies the logic of activation of each component (or node) of the system and aims at abstracting away quantitative aspects related to kinetics and molecule abundances. For instance, it may specify that component $c$ can turn on whenever its activator $b$ is on provided its inhibitor $a$ is off. Considering that the activity of components in the underlying system is not binary, the on and off actually relate to activity/abundance of molecules being above or below an interaction threshold. However, one may wonder whether such a binary coarse-graining may impede the validation of the model, leading to reject a BN although it describes the logic of components' activities correctly.

Figure 1 illustrates this issue with the incoherent feed-forward loop of type 3, I3-FFL[16], where an input node 1 directly inhibits the output 3, but indirectly activates it via node 2. The logic of nodes' activation is fixed: the activation of 3 requires that node 2 is sufficiently active and that node 1 is not sufficiently active. Theoretical studies with quantitative models[17,18] and experimental data from synthetically designed circuits[19] showed that, depending on kinetics parameters and starting from all nodes being inactive, a monotonic activation of the input can lead to a transient activity of the output (node 3). However, it is impossible to reproduce this behavior with usual (a)synchronous interpretations of BNs: starting from the state where all nodes are inactive, neither 2 nor 3 can be activated without the prior activation of 1. If 1 is active, 2 is active, but any transient activation of 3 is prevented (Fig. 1(d)).

Additional model features, such as intermediate levels for the nodes, or delays in interactions, would allow a transient activation for the I3-FFL output. However, such features come with additional parameters and higher computational cost, which limits their general application to large-scale networks.

This simple example seems to show that setting binary activities for nodes can both generate spurious behaviors (as expected with qualitative models), and also preclude the recapitulation of existing behaviors. Therefore, the validity of a model cannot be assessed by the usual (a)synchronous interpretations of BNs. This limitation largely impedes the inference of dynamical network

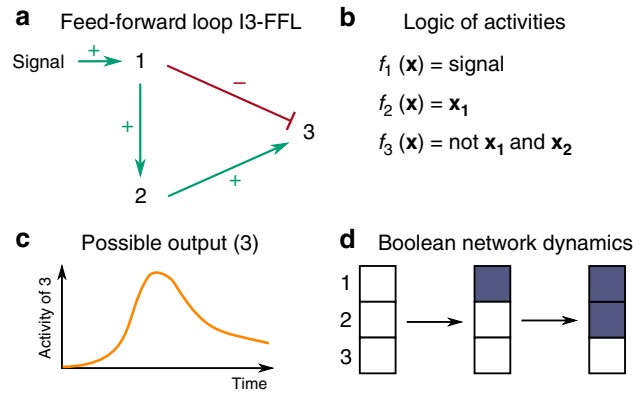

**Fig. 1 Boolean modeling of the incoherent feed-forward loop of type 3 (I3-FFL). a** the I3-FFL network and (**b**) its associated Boolean logic for nodes activities; $f_{1,2,3}(\mathbf{x})$ are the Boolean functions used to compute the next value of each node from a given configuration $\mathbf{x}$ of the network, which is here a binary vector specifying the current value of each node, $\mathbf{x}_i$ referring to the Boolean value of node $i$. Whereas theoretical and experimental studies showed that starting from all nodes being inactive, an activation of the output is possible when the signal is turned on (**c**), BN analysis cannot predict this transient behavior: (**d**) shows the corresponding complete dynamics of $f$ starting from the configuration where all nodes are inactive, and signal is set to 1. Configurations are represented by piles of three squares, where the top square represents the state of the first component, and so forth. A white square represents the inactive (0) state; a blue square represents the active (1) state; a dashed line indicates that no further evolution is possible. Arrows indicate possible transitions. The node 3 is never predicted to be active.

models and the identification of necessary interaction network motifs since the Boolean interpretation can wrongly conclude that no BN matching with a network motif can reproduce the desired behavior.

However, we found that this issue is actually due to the interpretations of BNs and not to their intrinsic Boolean nature. We introduce a simulation approach, the Most Permissive Boolean Networks (MPBNs), which presents the formal guarantee to capture all behaviors achievable without the need for additional parameters. If MPBNs cannot reproduce a given observation, no quantitative refinement of the Boolean model can do it, and the model can safely be considered as incoherent with the observations. While predicting more behaviors than synchronous and asynchronous interpretations of BNs, MPBNs still capture essential dynamical features of biological models.

Moreover, we demonstrate that the analysis of MPBNs avoids the state-space explosion problem, a strong limiting factor for the synchronous and asynchronous interpretations of BNs. The drastically reduced computational cost enables the precise qualitative analysis of dynamics of genome-scale networks.

## Results

**Preliminaries.** Computational modeling of dynamical systems relies on two fundamental ingredients: a language to specify the model, and an execution semantics. The language provides symbols and syntax rules to write a model, while the execution semantics mathematically defines how to interpret it. The semantics formalizes the notion of network configurations (or states) and how to compute their evolution over time. It provides an exhaustive assessment of model capabilities by enabling dynamical analyzes such as simulations as well as formal verification by invariant analysis and model-checking.

A BN is specified by a mathematical function mapping any binary vector of dimension $n$ to another binary vector of the same

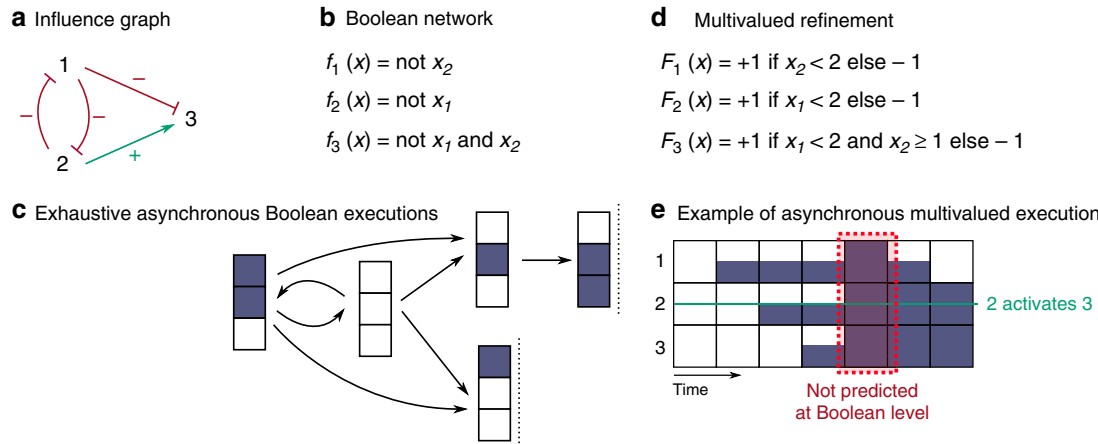

**Fig. 2 Example of qualitative models for the interactions between three components. a** Influence graph denoting the activation and inhibition relationships. **b** Example of a compatible BN, which defines the activation conditions of each component. **c** Exhaustive list of transitions obtained from the initial configuration where all three components are inactive. **d** Example of a multivalued network refining the BN (**b**) with components able to exist under three states (0, 1, 2). **e** Example of asynchronous execution of the multivalued network from the configuration 000. Half and fully blue squares represent the states 1 and 2.

dimension:

$$f : \mathbb{B}^n \to \mathbb{B}^n, \tag{1}$$

where $\mathbb{B} = \{0, 1\}$ represent the Boolean values. Each component of a binary vector models the state (inactive/active, absent/present) of the associated network node, and $f_i$ is the function which specifies the state towards which the $i$th component evolves. Figure 2(b) gives an example of a BN modeling a switch system.

BNs semantics computes the possible temporal evolutions of the component states using different methods. With synchronous executions of BNs (introduced by Kauffman[1]), we update all the components of the network at the same time, and a configuration $\mathbf{x} \in \mathbb{B}^n$ can only evolve to one configuration $f(\mathbf{x})$. With fully asynchronous executions of BNs (introduced by R. Thomas and usually referred to more simply as asynchronous in the computational systems biology literature), we update only one component at a given time, and a configuration $\mathbf{x} \in \mathbb{B}^n$ can evolve to any configuration which differs only by a single component $i$ where $f_i(\mathbf{x}) \neq \mathbf{x}_i$. This introduces potential non-determinism in the model trajectory since there can be different executions of the same BN from a given initial configuration. The (fully) asynchronous semantics is often described as more realistic for modeling biological networks, accounting for different kinetics of interactions.

Many more variants of executions of BNs have been studied in the literature, some imposing a precise order in the updating of the components, others allowing subsets of components to be updated simultaneously, etc. Most, if not all, generate a subset of the executions achievable with the (generalized) asynchronous semantics of BNs where any number of components can be updated at a time: a configuration can evolve to any other configuration that complies with the logical functions for the components that differ between both. Formally, for any $\mathbf{x}, \mathbf{y} \in \mathbb{B}^n$,

$$\mathbf{x} \xrightarrow[a]{f} \mathbf{y} \iff \forall i \in \Delta(\mathbf{x}, \mathbf{y}), \mathbf{y}_i = f_i(\mathbf{x}), \tag{2}$$

where $\Delta(\mathbf{x}, \mathbf{y})$ is the list of components which state differs between $\mathbf{x}$ and $\mathbf{y}$, i.e., $\Delta(\mathbf{x}, \mathbf{y}) = \{i \in \{1, \ldots, n\} | \mathbf{x}_i \neq \mathbf{y}_i\}$.

A configuration $\mathbf{y} \in \mathbb{B}^n$ is reachable from $\mathbf{x} \in \mathbb{B}^n$ if either $\mathbf{x} = \mathbf{y}$, or there exists a sequence of transitions from $\mathbf{x}$ to $\mathbf{y}$:

$$\rho_a^f(\mathbf{x}) = \left\{ \mathbf{y} \in \mathbb{B}^n | \mathbf{x} = \mathbf{y} \text{ or } \mathbf{x} \xrightarrow[a]{f} \cdots \xrightarrow[a]{f} \mathbf{y} \right\}. \tag{3}$$

Notice that if $\mathbf{y} \notin \rho_a^f(\mathbf{x})$, then it is impossible to evolve from $\mathbf{x}$ to $\mathbf{y}$ according to any of the semantics defined above, including the synchronous and fully asynchronous ones. Figure 2(c) shows all possible asynchronous evolutions of the example BN from the configuration where all the components are inactive, i.e. $\rho_a^f(000) = \{000, 110, 010, 011, 100\}$.

Reachability is a fundamental property to assess the compatibility of BN models with time series data: if none of the configurations matching an observation at a given time is reachable from any configuration matching an experimental observation at an earlier time, the BN cannot capture the observed behaviors.

Another prominent dynamical property studied with BNs, strongly linked to reachability, are attractors. Attractors represent the long-term behaviors of the model and are often used to represent cell phenotypes. Formally, an attractor is a smallest non-empty set of configurations from which it is impossible to escape: $A \subseteq \mathbb{B}^n$ is an attractor if and only each of its configuration $\mathbf{z} \in A$ verifies $\rho_a^f(\mathbf{z}) = A$. An attractor is said to be a fixed point whenever it is a single configuration $\mathbf{z} \in \mathbb{B}^n$ (whenever $f(\mathbf{z}) = \mathbf{z}$ with the asynchronous semantics), and complex if it is an ensemble of configurations, such as cyclic attractors, modeling potential sustained oscillations.

**Refinements of BNs**. BNs impose a drastic coarse-graining on component activity. Several modeling frameworks introduced a finer granularity in logical models[20]. Examples include Multi-valued Networks (MNs)[21], where components can take more than two logical values (0, 1, 2,…, $m$), fuzzy logic[22], which extends logical models with continuous domains, stochastic extensions of fully asynchronous BNs[23], and ordinary differential equations (ODEs)[24,25], where values of components are non-negative reals and vary along continuous time. Their specifications require, however, much more information about the biological system, such as thresholds of interactions for MNs and precise kinetics for ODEs. These parameters are often unknown, and their automatic inference would require a significant amount of data collected in similar experimental settings.

One could use any of these frameworks to model the same biological system at different abstraction levels. Which raises the question of the relationship between models from different frameworks: is a MN model $F$ a refinement of a BN model $f$? In

other words, does $F$ specify a system with more quantitative information than $f$, while following the same (Boolean) logic for the interactions.

We consider here a simple mathematical criterion for refinements: the value of a component can decrease (resp. increase) only if the component can be set to 0 (resp. 1) in the BN with a possible binarization of the state. A formal definition will be given in the next section.

**Incompleteness of (a)synchronous BNs.** We can define a MN $F$ of dimension $n$ by a discrete function, which maps, for each component, states to the tendency of value change (decrease, steady, increase). To ease notations, and without loss of generality, we assume that all the components can take an integer value between 0 and the same fixed $m$:

$$F : \mathbb{M}^n \to \{-1, 0, 1\}^n, \qquad (4)$$

where $\mathbb{M} = \{0, 1, \dots, m\}$. The successors of a configuration $\mathbf{x} \in \mathbb{M}^n$ are then computed by adding the value of $F(\mathbf{x})$ to (a subset of) components, provided they stay non-negative and do not exceed their maximum value $m$.

In Fig. 2 we present a simple example of BN for which asynchronous executions miss possible behaviors of the network when considering a multivalued refinement of it. The MN in Fig. 2(d) is a refinement of the BN in Fig. 2(b). In addition to the higher granularity for the activity levels of all three components, it brings additional information on the activation of component 3. An intermediate value of 2 is sufficient to activate 3 provided that the value of its inhibitor 1 is not high. One of its asynchronous execution shown in Fig. 2(e) predicts that the three components can get activated simultaneously, which was never predicted by any of the asynchronous executions of the BN. Assuming the validation of the model were subject to the reachability of a configuration with all the three components active from a configuration with all the components inactive, this BN model would be deemed insufficient for achieving the observed behavior, with an erroneous conclusion that its logic is wrong.

**The Most Permissive execution paradigm for BNs.** The critical reason usual BN interpretations miss behaviors is that the binary coarse-graining coupled with the instantaneous state changes preempt interactions occurring during the course of (de)activations. In the counter-example of Fig. 2, Boolean interpretations exclude the activation of component 3 during the activation of components 1 and 2, whereas, in a possible refinement, 3 can indeed increase before 2 reaches its fully active state and before 1 is sufficiently expressed to inhibit it.

We devised a dynamical interpretation of BNs, called Most Permissive semantics, in which we consider that a component can exist in 4 states: inactive (0), increasing (↗), decreasing (↘), or active (1). While a component is in a dynamic state (↗ or ↘), it can be read non-deterministically as either 0 or 1. These ambiguous states account for the absence of information on actual influence thresholds: a component in a dynamic state can be above the influence threshold for one component while being below the influence threshold for another one.

Figure 3 summarizes the changes of component states possible with the Most Permissive semantics. A component $i$ can change to the ↗ (resp. ↘) state from the 0 or ↘ (resp. 1 or ↗) state whenever it can interpret the value of its regulators in a way which makes its logical function $f_i$ true (resp. false) – if one of its regulators is in a dynamic state, both Boolean interpretations can be considered. Once in ↗ (resp. ↘) state, it can reach 1 (resp. 0) at any time. As a result, a component cannot go from ↗ (resp. ↘) state to 0 (resp. 1) without going through the ↘ (resp. ↗) state.

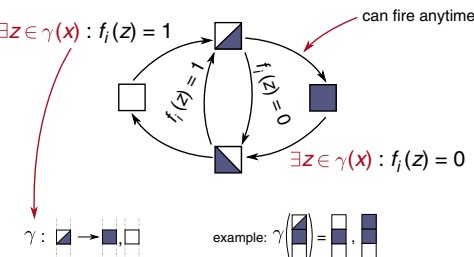

**Fig. 3 Allowed changes of component states in Most Permissive Boolean Networks.** Conditions are expressed for a component $i \in \{1, \dots, n\}$ from a most-permissive configuration $\mathbf{x} \in \{0, ↗, ↘, 1\}^n$. The increasing state ↗ is represented by a top-left white and bottom-right blue square, the decreasing state ↘ by a bottom-left blue and top-right white square. The function $\gamma$ gives the admissible Boolean interpretations of $\mathbf{x}$: $\gamma(\mathbf{x}) = \{\mathbf{z} \in \mathbb{B}^n | \forall i \in \{1, \dots, n\}, \mathbf{x}_i \in \mathbb{B} \Rightarrow \mathbf{z}_i = \mathbf{x}_i\}$, i.e., all the components in Boolean states are fixed, and the others are free.

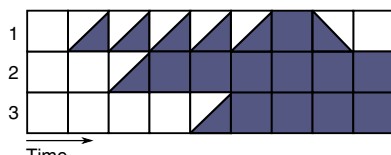

**Fig. 4 One of the possible executions using the Most Permissive semantics on the BN in Fig. 2(b).** The execution starts from the configuration where all components are inactive. Note that it correctly recovers the (transient) reachability of the configuration where the three components are active.

Each component evolves independently of all others. The complete formal definition is given in Supplementary Note 2.

Figure 4 shows an example of execution using the Most Permissive semantics on the BN of Fig. 2. Contrary to the (a)synchronous interpretations, the Most Permissive semantics correctly captures the possible (transient) reachability of the configuration where the three genes are active. While component 1 is ↗ and component 2 is active, gene 3 can indeed change to ↗, thus leading to the activation of all three components. This configuration is not in an attractor, and both single-point attractors identified in Fig. 2(c) are reachable via different Most Permissive executions.

We provide in Supplementary Fig. 1 the application of MPBNs to the BN of the I3-FFL motif presented in Fig. 1, which successfully captures the transient activation of node 3. Even if we allow changing the Boolean logic, it is the only BN that can reproduce the observed transient and steady behaviors (Supplementary Note 3.D). Therefore, a Boolean asynchronous analysis would have concluded that the network motif is insufficient to reproduce the observed behavior.

**Formal guarantees for model refinements.** Using the simple examples in Figs. 1 and 2, we have shown that BN refinements can introduce behaviors that cannot be captured with classical semantics.

MPBNs bring the formal guarantee of being able to reproduce all the behaviors achievable in any refinements, being a MN or an ODE system (Theorem 1 and Corollary 1 in Supplementary Note 2). In other words, if the Most Permissive semantics concludes that it is impossible to observe a given state change for some components, then no qualitative or quantitative model verifying the refinement criteria can predict these state changes.

The refinement criterion relies on a binarization of the multivalued configuration. An appropriate binarization necessarily

quantifies 0 as Boolean 0 and $m$ as 1, and is free for the other intermediate values. Let us denote by $\beta(\mathbf{x})$ the set of possible binarization of configuration $\mathbf{x} \in \mathbb{M}^n$:

$$\beta(\mathbf{x}) = \{\mathbf{x}' \in \mathbb{B}^n | \forall i \in \{1, \dots, n\}, \mathbf{x}_i = 0 \Rightarrow \mathbf{x}'_i = 0$$
$$\text{and } \mathbf{x}_i = m \Rightarrow \mathbf{x}'_i = 1\}. \quad (5)$$

For example with $m = 2$, $\beta(012) = \{001, 011\}$.

Then, we say a MN $F$ is a refinement of a BN $f$ of the same dimension $n$ if and only if for every configuration $\mathbf{x} \in \mathbb{M}^n$, and for every component $i \in \{1, \dots, n\}$, $F_i(\mathbf{x}) < 0$ implies there exists $\mathbf{x}' \in \beta(\mathbf{x})$ such that $f_i(\mathbf{x}') = 0$, and $F_i(\mathbf{x}) > 0$ implies there exists $\mathbf{x}' \in \beta(\mathbf{x})$ such that $f_i(\mathbf{x}') = 1$.

This characterization of BN refinement to MN can be directly extended to ODEs. Indeed, ODEs specify the (real) derivative of the (positive real) value of each component:

$$\mathcal{F} : \mathbb{R}^n_{\geq 0} \to \mathbb{R}^n . \quad (6)$$

Only the binarization $\beta$ should be adapted in (5) to reflect that there is no (a priori) upper bounded value $m$ for components.

The completeness property states the following. Consider a multivalued refinement $F$ of a BN $f$ with which there exists an asynchronous trajectory from a multivalued configuration $\mathbf{x}$ to $\mathbf{y}$. Let us write $\hat{\mathbf{x}}$ any most-permissive configuration compatible with $\mathbf{x}$: if $\mathbf{x}_i = 0$, then $\hat{\mathbf{x}}_i = 0$, if $\mathbf{x}_i$ is the maximum value of $i$, then $\hat{\mathbf{x}}_i = 1$, and in the other cases $\hat{\mathbf{x}}_i$ can be either $\nearrow$ or $\searrow$. Then, there exists a most-permissive trajectory leading from any of these $\hat{\mathbf{x}}$ to a most-permissive configuration $\hat{\mathbf{y}}$ compatible with $\mathbf{y}$ and which is consistent with the the changes between $\mathbf{x}$ and $\mathbf{y}$: $\hat{\mathbf{y}}_i = \nearrow$ if $\mathbf{y}_i > \mathbf{x}_i$ and $\mathbf{y}_i < m$, $\hat{\mathbf{y}}_i = \searrow$ if $\mathbf{y}_i < \mathbf{x}_i$ and $\mathbf{y}_i > 0$, and $\hat{\mathbf{y}}_i = \hat{\mathbf{x}}_i$ if $\mathbf{y}_i = \mathbf{x}_i$. As the proof relies solely on the sign of the derivative of the refinement of $f$, the property extends to ODE refinements, which can be seen as multivalued networks with $m$ to infinity.

Allowing any state change without restriction would also provide the above guarantee. It appears that if there is a most-permissive trajectory between two binary configurations, then there is a multivalued refinement of the BN showing an asynchronous trajectory between matching multivalued configurations (Theorem 2 in Supplementary Note 2). Therefore, the completeness property can be achieved only by predicting at least the behaviors of MPBNs. In other words, the Most Permissive semantics is the tightest Boolean abstraction of multivalued refinements regarding reachability properties.

**Simpler computational complexity**. Most computational analyzes of BNs focus on two elementary dynamical properties: the reachability, which is the existence of a trajectory between two given configurations, and the existence of attractors. Here, we study these properties in term of algorithmic complexity classes. These theoretical results have very concrete implications for the analysis of MPBNs, making the approach scalable to genome-scale networks.

We first recall the bases of computational complexity classes[26]: the P class is formed by the algorithms running in time polynomial with the size of its inputs; the NP class by the algorithms running in polynomial time with non-deterministic choices; the PSPACE class by the algorithms running in polynomial space. We know that $P \subseteq NP \subseteq PSPACE$, where "$\subseteq$" can be understood as "simpler". A problem is complete for a given complexity class if it belongs to and is among the hardest problems of this class. The famous SAT problem of determining if a formula expressed in propositional logic (essentially Boolean variables and logic connectors) has a satisfying solution is NP-complete. It is not known yet if NP = PSPACE, but in practice, NP-complete problems are much more tractable than PSPACE-

complete ones, by several orders of magnitude. Hereafter, we also refer to the coNP class, delimiting the problems for which finding a counter-example is in NP, and to the $P^{NP}$ and $coNP^{coNP}$ classes, where $A^B$ denotes the problems that can be solved with complexity A assuming problems of class B can be solved in one instruction (oracle); note that $P^P = P$ and $NP^P = NP$. These complexity classes belong to the polynomial hierarchy, and are subject to the following properties: $NP \subseteq P^{NP}$ and $coNP \subseteq P^{NP} \subseteq coNP^{coNP} \subseteq PSPACE$.

With asynchronous BNs, it is challenging to determine if a trajectory exists between two configurations since, in the worst case, it requires exploring all the possible configurations. With MPBNs, this problem is much simpler thanks to an intriguing property: if there exists a trajectory between two configurations, then there is such a trajectory visiting at most $3n$ configurations. Intuitively, this shortcut corresponds to a particular sequence of state changes: in a first phase, only transitions changing a state from 0 or 1 to $\nearrow$ or $\searrow$ take place; In a second phase, only transitions changing states within $\nearrow$ and $\searrow$; in a final phase, only transitions changing states from $\nearrow$ or $\searrow$ to 1 or 0. Each phase comprises at most $n$ transitions, one for each component.

Moreover, finding this shortcut requires exploring at most a quadratic number of transitions in the general case, and only $3n$ whenever the target configuration is in an attractor. The exploration consists of performing as many transitions as possible of the first phase, putting the largest possible number of components in a dynamic state. For each component whose state does not change between the starting and target configuration, it is then necessary to switch the dynamic state back (second phase). If this is not possible, then the exploration is repeated from the beginning while preventing this specific component from changing to a dynamic state (as it would still be impossible to go back to the initial binary state, and the target configuration would not be part of an attractor). Overall, the exploration is thus repeated at most $n$ times. Finally, all the transitions of the third phase are applied, which should lead to the target configuration if and only if it is reachable.

On the other hand, determining the possibility of a most-permissive transition is NP-complete in the general case: indeed, the condition "$\exists \mathbf{z} \in \gamma(\mathbf{x}): f_i(\mathbf{z}) = 1$" in Fig. 3 is the SAT problem. For biological networks, it is usual to assume that components cannot have both positive (activator) and negative (inhibitor) direct influences. The resulting BNs are called locally monotonic: each local function $f_i$ is monotonic for every component it depends on: increasing the number of activators (resp. inhibitors) of $i$ in state 1 can only increase (resp. decrease) the value of $f_i$. Thus, determining the existence of a Boolean interpretation $\mathbf{z}$ of a most-permissive configuration $\mathbf{x}$ so that $f_i(\mathbf{z}) = 1$ comes down to considering activators in dynamic state as 1 and inhibitors in dynamic state as 0, and conversely for $f_i(\mathbf{z}) = 0$. Therefore, determining the possibility of a most-permissive transition can be done in linear time with locally monotonic BNs.

The reachability problem in MPBNs can thus be solved in polynomial time whenever $f$ is locally monotonic (Theorem 3 in Supplementary Note 2), a considerable drop in complexity compared to synchronous or asynchronous BNs where the problem is PSPACE-complete (Supplementary Note 1). With non-locally monotonic BNs, the reachability problem is in $P^{NP}$.

While the attractors of asynchronous BNs can be complex objects, the attractors of MPBNs are particular mathematical objects called minimal trap spaces. A trap space is a hypercube which is closed by $f$: for any vertex $\mathbf{x}$, $f(\mathbf{x})$ is also a vertex. A trap space is minimal whenever it does not include a different trap space. Attractors in MPBNs have this regular structure because whenever two configurations lying on any diagonal of an

hypercube are reachable from each other, they can reach the adjacent configurations as well.

Determining if a configuration $\mathbf{x} \in \mathbb{B}^n$ belongs to an attractor of $f$ is a key problem to identify attractors of a BNs. It is again a PSPACE-complete problem for synchronous and asynchronous BNs (Supplementary Note 1). In the case of MPBNs, it boils down to verifying if the trap space containing $\mathbf{x}$ is minimal, which is at most of complexity coNP for locally monotonic BNs, and at most coNP$^{coNP}$ for non-locally monotonic BNs (Theorem 4 in Supplementary Note 2). The computation of minimal trap spaces of a BN can be performed efficiently with SAT solvers and related logic programming frameworks[27].

Finally, notice that determining a configuration $\mathbf{x} \in \mathbb{B}^n$ which both belongs to an attractor of $f$ and which is reachable from another configuration $\mathbf{y} \in \mathbb{B}^n$ has the same complexity as the attractor problem: it is PSPACE-complete with synchronous and asynchronous BNs, whereas it is at most coNP for locally monotonic BNs and coNP$^{coNP}$ for non-locally monotonic BNs. Therefore, MPBNs offer a drastic reduction in the theoretical computational complexity for analyzing reachability, attractors, and reachable attractors of BNs, with practical implications in term of scalability of Boolean modeling: on a regular 3.3GHz processor, our implementation of MPBNs can compute reachable attractors of randomly generated scale-free networks[28] with 1000 components in a fraction of a second, less than 2 s with 10,000 components, and less than 50 s with 100,000 components (Supplementary Note 3.C, Supplementary Fig. 4).

**Validation of MPBNs on actual biological models**. An essential feature of logical models is their ability to conclude on the absence of certain behaviors. For instance, differentiation processes are modeled using separate attractors representing the final phenotypes and trajectories where configurations are committed to reaching a particular attractor with no possibility to rejoin other differentiation branches. A model allowing any configuration to reach any attractor would indeed be useless without quantitative aspects. We will show that, although enabling more behaviors than (a)synchronous BNs, MPBNs are still constraining and able to capture differentiation and cell fate decisions.

As we have said above, attractors of MPBNs correspond to the minimal trap spaces of the Boolean function. Prior work has shown that these trap spaces match well with the complex attractors of fully asynchronous BNs in many real-world models of biological networks[27]. We illustrate this in Supplementary Note 3, with the computations of attractors in a logical model of bladder tumorignesis[29], where attractors match with asynchronous BNs and are computed in milliseconds, while taking several minutes and even time out with asynchronous simulation methods[30].

To further assess the predictive capacity of MPBNs in practice, we reproduced studies on logical models of differentiation which involve delineating the set of attractors reachable from different initial conditions (Supplementary Note 3). Due to the formal guarantees of MPBNs stated in previous sections, MPBNs will recover at least the reachable states identified using asynchronous analysis. In the following case studies, despite predicting potentially more behaviors, MPBNs rules out the same set of attractors that the asynchronous analysis, and at a much lower computational cost.

In the case of a tumor invasion model[8] (Supplementary Fig. 2), all the attractors are fixed points, and thus are identical in MPBNs. The study focused on the reachability of these attractors from a set of initial conditions with different combinations of mutants. One of the main prediction is the synergistic combination of p53 loss of function and Notch gain of function which lead

to the loss of reachability of attractors corresponding to cell death. The MPBN analysis recovers the exact same set of reachable attractors with the different combinations of mutations than reported with fully asynchronous analysis.

In the case of T-cell differentiation[9] (Supplementary Fig. 3), the study focused on identifying changes of input conditions which trigger a change of attractor, resulting in a reprogramming graph across pre-determined T-cell subtypes. Due to the large size of the model (101 components), the original study had to perform approximations through model reduction and symbolic model-checking techniques, avoiding the need for computing attractors. On the other hand, MPBNs can efficiently handle the booleanized[31] original large multivalued model, list the attractors and compute their reachability following the input condition changes. The attractor computation enables determining that in most conditions the attractors are fixed points (and thus are identical in asynchronous BNs), in two conditions (APC and proTh1), the MPBN has one complex attractor, indicating the existence of at least one complex asynchronous attractor. Then, the Most Permissive reachability analysis concludes on the same reprogramming graph, at much lower computational cost.

In conclusion, as stated in previous sections, MPBNs are formally guaranteed to capture behaviors that only multivalued discrete models could capture with (a)synchronous interpretations; and as supported by these case studies, the Most Permissive interpretation of BNs is still stringent enough to capture processes that control reachable attractors, and doing so at a much lower computational cost.

## Discussion

The choice of the dynamical interpretation of BNs has drastic effects on their predictions. Whereas the (fully) asynchronous BN interpretation is often advised for practical applications, it overlooks behaviors emerging from different timescales for the interactions, leading to biases when selecting plausible models. Such misses are due to artifacts of configurations updates. On the contrary, MPBNs offer a framework for reasoning on the qualitative dynamics without making any strong a priori hypothesis about the timescale and thresholds of interactions, and without additional parameter.

The state-space explosion triggered by the synchronous and asynchronous interpretations of BNs is another significant bottleneck for their application in systems biology[3,32]. MPBNs offer drastic gains in computational complexity when analyzing possible trajectories and attractors, both elementary and essential properties, underpining the potential of a model. In practice, the verification of these properties with asynchronous BNs is typically limited to networks with 50–100 nodes. In contrast, deciding the reachability and attractor properties in MPBNs relies on scalable algorithms and does not suffer from the state-space explosion. For the case of locally monotonic BNs, which is a classical hypothesis for biological networks, the complexity allows addressing very large-scale networks, as illustrated in Supplementary Note 3, with experiments on BNs with up to 100,000 components.

The prediction of attractors reachable from specific initial conditions, and possibly under various mutant conditions, is at the core of many studies using logical models. While MPBNs can identify the complete set of reachable attractors several orders of magnitude faster than asynchronous BNs, the quantification of the propensities of each attractor, e.g., performed by sampling the trajectories[23,30], is yet to be explored. In addition to the validation and the control of genome-scale models, the complexity breakthrough brought by MPBNs together with their ability to overcome artifacts of Boolean modeling paves the way

towards the inference and learning of large-scale logical models from experimental data.

**Reporting summary**. Further information on research design is available in the Nature Research Reporting Summary linked to this article.

## Data availability
Notebooks for reproducing the case studies are available at https://doi.org/10.5281/zenodo.3936123, with instructions for their execution.

## Code availability
Our software tool mpbn implementing reachability and attractor analysis in BNs with Most Permissive semantics is available at https://github.com/pauleve/mpbn and https://doi.org/10.5281/zenodo.3946585, and is integrated in the CoLoMoTo notebook environment[33] available at http://colomoto.org/notebook.

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

## Acknowledgements
Research was supported by the French Agence Nationale pour la Recherche (ANR) in the context of ANR-FNR project AlgoReCell (ANR-16-CE12-0034). The authors thank aSciStance Ltd. for their scientific advices and editing services.

## Author contributions
L.P., T.C., S.H. designed the research; L.P. defined MPBNs and demonstrated Theorems 1, 3, 4, implemented code, performed experiments, and initially drafted the manuscript; J.K. demonstrated Theorem 2; L.P., J.K., T.C., S.H. wrote the manuscript.

## Competing Interests
The authors declare no competing interests.
