## [Peer Review File · Nature Communications]

Reviewers' Comments:

Reviewer #1:

Remarks to the Author:

This manuscript introduces a new paradigm for qualitative modeling of biological systems: the Most Permissive Boolean Networks (MPBNs). The authors show that this type of model is guaranteed not to miss any behavior achievable by a quantitative model that follows the same logic. They also show that the question of whether a state is reachable from another state can be answered in a much more computationally efficient manner in this framework. Both of these are very valuable features which will be of great interest to the modeling and systems biology community. There are a number of points which would need to be clarified or rephrased in order to improve the understanding of the manuscript.

Major comments

1. The manuscript focuses on reachability-related questions. Yet, there are other important questions that Boolean models aim to answer, for example determining the complete attractor repertoire of a system, and determining the basins of attraction of each attractor. Would these questions also be easier to answer with Most Permissive Boolean Networks, or would these still be done in the traditional way, and the MPBNs would only be used to determine reachability?
2. Figure 1 gives an example where a traditional Boolean model (whether updated synchronously or updating one node at a time) would miss an observed non-monotonic behavior of a system component. Yet, the exact behavior is incompletely specified, and thus the example is not completely convincing. If one started the Boolean model from the initial condition $x_1=0$, $x_2=1$, $x_3=0$, it would be possible for x_3 to increase to 1 and then to decrease, thus qualitatively reproducing the trajectory in panel c. Why is only the initial state 000 considered in the Boolean model?
3. The presentation of the two applications is brief and not sufficiently informative. The Supporting Information has a single paragraph on biological applications. "Correctly predicted the loss of reachability of apoptotic attractors upon the mutations of p53 and NICD presented in the study (Fig. S2)" – this is reproduction of a single finding from the reference; Fig. S2 reproduces the influence graph of the reference. "In the case of T-cell differentiation (9), MPBNs recovered the same reprogramming graph between T-cell types (Fig. S3)" Fig. S3 the influence graph of the reference, not the reprogramming graph. The reader needs to understand all the ways in which the MPBN could go beyond the previous results.
4. There should be an illustration of MPBNs used to determine the attractors reachable from a given initial condition.

More minor points:

5. Several statements made in the manuscript should be supported by references or clarified. For example "... there is no guarantee that their analysis can be relevant for a more precise model, and thus for the actual biological system". Perhaps here the intended meaning is that there is no guarantee that all the results of a Boolean model can be reproduced by a quantitative model. One can agree with that sentence. But since a Boolean model is constructed to recapitulate the behavior of an actual biological system, and it is validated by comparing to the actual behavior, a validated Boolean model is relevant for the actual biological system. Yes, it can happen that one of the model's predictions is tested and invalidated. If this happens, one of the model's assumptions is refined in such a way that the previous agreements are maintained. This is doable because there is never enough information to completely specify the regulatory functions. Do the authors know of any specific cases where a model showed spurious behavior that could not be corrected? If so, they should cite them.

Another example is on line 58 "This limitation largely impedes the inference of dynamical network models and the identification of necessary interaction network motifs since the Boolean interpretation can strongly distort the landscape of candidate models." References are needed to support both parts of this statement.

In line 439 "MPBNs are formally guaranteed never to ignore behaviors hidden by artifacts of usual Boolean modeling". Is the intended meaning "formally guaranteed to capture behaviors that only multi-level discrete models could capture before"?

6. What does "*" stand for in Equation [3]?

7. What does $P^{\{NP\}}$ mean in line 331?

8. In many instances the Supporting information has question marks instead of references.

9. Suggested word changes:

Line 53 "discovery" -> recapitulation

Line 140 "predict" -> capture

Reviewer #2:

Remarks to the Author:

Dear Editor and Authors,

In ms 20-12959, the authors introduce an extension to the Boolean Network (BN) framework, where instead of nodes existing as ON or OFF (1 or 0), they can exist in intermediate "increasing" or "decreasing" states. Combined with an appropriate asynchronous update scheme, the authors show that their model captures traditional BN dynamics but can be analyzed with less computational complexity than BNs, which in the worst-case scenario requires analysis of the complete state transition network (with 2^n states for a n-component BN). I found this to be an interesting result that will likely be of significant interest to researchers in the network science and/or systems biology communities.

The authors also argue that their framework successfully captures dynamic behavior that is not accessible to BN without significantly increasing its complexity, e.g. by moving to a multi-state framework and necessarily increasing the size of the state transition network, e.g. from 2^n to 3^n . I do not disagree with the authors in their conclusions, but their arguments here are not as crisp as they could be because they occasionally conflate the "influence graph" (e.g. "A has some sort of positive influence of B") with the Boolean rules that define the network (e.g. " $f_B = A$ or C "); see comments 2 and 4, below. Elsewhere, the authors risk losing some readers early in the paper with some occasionally broad language regarding the limitations of BNs (comment 1). Finally, some additional details of the authors' model would clarify the paper (comment 5).

I commend the authors for an interesting study and I hope my detailed comments, below, will be useful as they consider revisions.

Comment 1

On pg1 col1 the authors write, "Boolean Networks are often created from scratch..." I take their meaning here, but it seems to me that more care is warranted here. Some readers could understandably infer the present language to mean "made up without any experimental evidence", which is certainly not what the authors intend to convey.

Similarly, in the same paragraph the authors write, "...there is no guarantee that their analysis can be relevant for a more precise model..." It seems to me that a successful BN aims to capture the broad behavior of the system being modeled. While it is certainly the case that a BN will not capture finer details, and that spurious predictions can occur, it seems to me that suggesting a strong BN is not relevant without qualification is perhaps too strong.

Comment 2

Fig. 1d is identified in the caption as showing the "complete dynamics of f ", but it seems to me this is not correct for two reasons. First, in a 3-node BN there are $2^3 = 8$ states, whereas the authors here show 3. Second, The first transition, from state 000 to state 100, can only occur based on a changing input signal rather than from the internal dynamics dictated by the update rules. It seems more natural here to show the complete state transition network. If desired one could indicate (with a different type of edge, perhaps) transitions that can occur by forcing the input signal to change (i.e. switching the state of node 1 from 0 to 1 or vice versa) to distinguish it from the natural dynamics of the system.

Moreover, I don't think the authors are making a particularly strong case against BNs here; to my eye the takeaway from this example is "this BN does not capture the experimental behavior of the system", which simply suggests a different BN should be chosen. (Indeed, this neatly describes a common situation that arises when BNs are developed.) To be compelling here the authors need to show that -no- BN can show transient activation (which is not possible, see below) or shift their argument to make clear that the "ON vs. OFF" nature of BNs inherently fails to capture the detail of a continuous model.

A three-node network that does capture transient activation is

$f_1 = \text{signal (fix to ON)}$

$f_2 = x_1$

$f_3 = \text{not}(x_1 \text{ and } x_2)$

In a synchronous update scheme, the following sequence of states can be observed:

100 -> 111 -> 110

In short, it seems to me that Fig. 1 and the main text pertaining to it (through the beginning of pg2) should be revised.

Comment 3

On pg2 col1 the authors write, "Therefore, the validity of a model cannot be assessed by the usual interpretations of BNs." The meaning of "usual interpretations" here (and in the following paragraph) should be clarified.

Comment 4

On pg3 col2 the authors write, "with an erroneous conclusion that its underlying influence graph is wrong" when discussing the failure of a BN to capture observed dynamic behavior (Fig. 2). This does not seem correct to me; a BN is not fully defined by its influence graph (Fig. 2a) but rather by its update rules (Fig. 2b). Similar to my comment 2, it seems to me here that one would first search for a different parameterization of the influence graph that succeeds in capturing the observed behavior.

Comment 5

On pg4 col1, the authors write, "A component i can change to the increasing (resp. decreasing)..." and then later "Once in increasing (resp. decreasing) state, it can reach 1 (resp. 0) at any time." This second quote makes it clear that the first quote is referring to changing from state 0 to increasing and changing from state 1 to decreasing. I suggest changing the first quote to make this explicit.

Indeed, it might be useful to indicate near this text what transitions are possible between the 4 states (0, inc, dec, 1), as the text does not address whether it is possible to go back from increasing to 0 and from decreasing back to 1. (A table might work well.)

Moving to the "derivative of a continuous model" framework, one could argue that you start by going from 1 (a maximum) to decreasing and proceed by moving back to 1 without first passing through a minimum (0) and then a region with a positive slope (increasing). However, in the Boolean reduction one could consider such fluctuations occurring uniformly above or below the activation/deactivation threshold. Thus some interpretation here would be useful.

Comment 6

On pg5 col1 the authors introduce some notation (e.g. $P^{\{NP\}}$), which should be defined for readers who are unfamiliar with it.

Comment 7

In the SI it appears that several Latex references are broken (e.g. the beginning of section 3).

Response letter

We are grateful to the reviewers for their positive feedback and suggestions for strengthening the presentation of the results. Besides improvements on the phrasing of different statements, this revision brings the following main changes:

- We clarified the introductory example settings, by better justifying the choice of 1) initial conditions, 2) observations coming from theoretical analysis of quantitative models, and 3) experimental data from synthetically-designed circuits;
- We present a short analysis showing that no (a)synchronous BN exists that respect the I3-FFL motif and the corresponding observed behaviors, strengthening the points around network motif validation which can be hindered by the pure Boolean (a)synchronous interpretation;
- We present a more detailed description of the analyses performed for the case studies to better emphasize the computational gain when computing reachable attractors, while MPBNs remain stringent enough to reproduce previous predictions.

Please see below our point-by-point response to each reviewer. A PDF tracking the changes made in the text is enclosed.

Point-by-point response

Reviewer #1

1. The manuscript focuses on reachability-related questions. Yet, there are other important questions that Boolean models aim to answer, for example determining the complete attractor repertoire of a system, and determining the basins of attraction of each attractor. Would these questions also be easier to answer with Most Permissive Boolean Networks, or would these still be done in the traditional way, and the MPBNs would only be used to determine reachability?

The complexity gain also applies to attractors and reachable attractors. Actually, a large part of the case studies on biological and randomly generated network involved the computation of reachable attractors and showed an improvement of computation time by several orders of magnitude. Determining whether a configuration belongs to the basin of a given attractor also boils down to the attractor reachability problem.

We clarified these results in several places in the revised manuscript: in the complexity section (p5, lines 428-442), where we added a paragraph to address the reachability of attractors, and in the case studies section (p6, lines 465-474), where we made explicit the fact that case studies rely on the determination of attractors and their reachability.

2. Figure 1 gives an example where a traditional Boolean model (whether updated synchronously or updating one node at a time) would miss an observed non-monotonic behavior of a system component. Yet, the exact behavior is incompletely specified, and thus the example is not completely convincing. If one started the Boolean model from the initial condition $x_1=0$, $x_2=1$, $x_3=0$, it would be

possible for x_3 to increase to 1 and then to decrease, thus qualitatively reproducing the trajectory in panel c. Why is only the initial state 000 considered in the Boolean model?

The initial state 000 is dictated by the theoretical and experimental studies of the I3-FFL network, showing in both cases that, starting from all node being inactive, an increase of the signal can lead (under some range of kinetics parameters) to transient activation of the output.

We updated the text in the introduction to clarify this point (lines 48-53).

3. The presentation of the two applications is brief and not sufficiently informative. The Supporting Information has a single paragraph on biological applications. "Correctly predicted the loss of reachability of apoptotic attractors upon the mutations of p53 and NICD presented in the study (Fig. S2)" – this is reproduction of a single finding from the reference; Fig. S2 reproduces the influence graph of the reference. "In the case of T-cell differentiation (9), MPBNs recovered the same reprogramming graph between T-cell types (Fig. S3)" Fig. S3 the influence graph of the reference, not the reprogramming graph. The reader needs to understand all the ways in which the MPBN could go beyond the previous results.

We improved the description of the case studies in the main text and SI to clarify the properties we can compute using MPBNs. These case studies show that, despite predicting more trajectories, MPBNs remain stringent enough to recover predictions made in prior work, that relied on the absence of certain attractors' reachability. Moreover, this section underlines the low computational cost of MPBNs analyses, allowing, for instance, to perform an explicit analysis of the attractors in the large-scale T-cell model, not performed in the original study.

Combined with the introductory example where we show how MPBNs can predict trajectories missed with asynchronous analyses, we believe this gives an accurate global picture of the improvement brought by MPBNs.

4. There should be an illustration of MPBNs used to determine the attractors reachable from a given initial condition.

Two of the case studies on models from the literature and the case study on very large random BNs actually relied on computing reachable attractors from specific initial conditions.

We improved the description of these case studies to make it explicit (p6, lines 465-502)

5. Several statements made in the manuscript should be supported by references or clarified. For example "... there is no guarantee that their analysis can be relevant for a more precise model, and thus for the actual biological system". Perhaps here the intended meaning is that there is no guarantee that all the results of a Boolean model can be reproduced by a quantitative model. One can agree with that sentence. But since a Boolean model is constructed to recapitulate the behavior of an actual biological system, and it is validated by comparing to the actual behavior, a validated Boolean model is relevant for the actual biological system. Yes, it can happen that one of the model's predictions is tested and invalidated. If this happens, one of the model's assumptions is refined in such a way that the previous agreements are maintained. This is doable because there is never enough information to completely specify the regulatory functions. Do the authors know of any specific cases where a model showed spurious behavior that could not be corrected? If so, they should cite them.

We aimed to describe the benefit of having a qualitative model insofar that we can directly transpose the result of its analysis to one of its quantitative refinements: if one predicts that x cannot reach y with asynchronous BNs, it remains possible that refining the model (e.g., by introducing intermediate levels) actually uncovers the existence of such a trajectory. With MPBNs, we have the guarantee that the Boolean analysis is complete.

We have completely rewritten this paragraph to explain this notion better (lines 31-42)

Another example is on line 58 “This limitation largely impedes the inference of dynamical network models and the identification of necessary interaction network motifs since the Boolean interpretation can strongly distort the landscape of candidate models.” References are needed to support both parts of this statement.

This statement comes from the fact that an (a)synchronous analysis can rule out a BN which actually encodes the correct logic of activations (as it is the case with the introductory example): if there exists no other BN compatible with the network motif and the expected dynamics, the analysis will conclude that the motif is not sufficient to reproduce the behavior.

We reworded this sentence to clarify this idea (lines 68-72)

Moreover, we added an analysis of the I3-FFL network motif demonstrating this issue: it turns out that the BN used in the introductory example is the only one matching the network motif and can reproduce the behavior observed with quantitative models and synthetically-designed circuits. Therefore, in this case, an (a)synchronous analysis would mis-conclude that the network is not sufficient. We included a new paragraph (p4, lines 265-272) explaining this result.

In line 439 “MPBNs are formally guaranteed never to ignore behaviors hidden by artifacts of usual Boolean modeling”. Is the intended meaning “formally guaranteed to capture behaviors that only multi-level discrete models could capture before”?

We thank the reviewer for this suggestion which we adopted.

6. What does “ \rightarrow^* ” stand for in Equation [3]?

The notation “ \rightarrow^* ” referred to a (possibly empty) sequence of transitions. We modified [3] to remove this abbreviation.

7. What does P^{NP} mean in line 331?

This complexity class denotes problems that can be solved in polynomial time, assuming problems of class NP can be solved in one instruction (known as an oracle). P^{NP} is known to be harder than NP, but simpler than PSPACE, i.e., simpler than reachability and attractors with (a)synchronous BNs.

We modified the relevant paragraph to improve the explanation of this complexity class (p5, lines 352-357).

8. In many instances the Supporting information has question marks instead of references.

We apologize for this mistake of compilation of the LaTeX document.

9. Suggested word changes:

Line 53 "discovery" → recapitulation

Line 140 "predict" → capture

We adopted these suggestions.

Reviewer #2

Comment 1

On pg1 col1 the authors write, "Boolean Networks are often created from scratch..." I take their meaning here, but it seems to me that more care is warranted here. Some readers could understandably infer the present language to mean "made up without any experimental evidence", which is certainly not what the authors intend to convey.

Similarly, in the same paragraph the authors write, "...there is no guarantee that their analysis can be relevant for a more precise model..." It seems to me that a successful BN aims to capture the broad behavior of the system being modeled. While it is certainly the case that a BN will not capture finer details, and that spurious predictions can occur, it seems to me that suggesting a strong BN is not relevant without qualification is perhaps too strong.

We aimed to describe the benefit of having a qualitative model insofar that we can directly transpose the result of its analysis to one of its quantitative refinements: if one predicts that x cannot reach y with asynchronous BNs, it remains possible that refining the model (e.g., by introducing intermediate levels) actually uncovers the existence of such a trajectory. With MPBNs, we have the guarantee that the Boolean analysis is complete.

We have completely rewritten this paragraph to explain this notion better (lines 31-42)

Comment 2

Fig. 1d is identified in the caption as showing the "complete dynamics of f ", but it seems to me this is not correct for two reasons. First, in a 3-node BN there are $2^3 = 8$ states, whereas the authors here show 3. Second, The first transition, from state 000 to state 100, can only occur based on a changing input signal rather than from the internal dynamics dictated by the update rules. It seems more natural here to show the complete state transition network. If desired one could indicate (with a different type of edge, perhaps) transitions that can occur by forcing the input signal to change (i.e. switching the state of node 1 from 0 to 1 or vice versa) to distinguish it from the natural dynamics of the system.

Moreover, I don't think the authors are making a particularly strong case against BNs here; to my eye the takeaway from this example is "this BN does not capture the experimental behavior of the system", which simply suggests a different BN should be chosen. (Indeed, this neatly describes a common situation that arises when BNs are developed.) To be compelling here the authors need to show that -no- BN can show transient activation (which is not possible, see below) or shift their argument to make clear that the "ON vs. OFF" nature of BNs inherently fails to capture the detail of a continuous model.

A three-node network that does capture transient activation is

$f_1 = \text{signal (fix to ON)}$

$f_2 = x_1$

$f_3 = \text{not } (x_1 \text{ and } x_2)$

In a synchronous update scheme, the following sequence of states can be observed:

100 -> 111 -> 110

In short, it seems to me that Fig. 1 and the main text pertaining to it (through the beginning of pg2) should be revised.

The I3-FFL network possesses a fixed logic, and we focussed on behaviors observed from a system where all nodes are initially inactive. Theoretical and experimental studies showed a transient activation of the output with this logic and initial state. The logic of the BN and its initial configuration reflect these settings and are correct with respect to the system: the theoretical studies used quantitative models, and the experimental study relied on synthetically-designed circuits, both reflecting the I3-FFL logic. Under some range of kinetic parameters and starting with all node inactive, they observed the transient activation of node 3 when increasing the signal.

The *complete* aspect was implicitly subject to *starting from 000*.

We updated the text and the legend of Fig. 1 to clarify these points (lines 43-57).

Moreover, we added an analysis of the I3-FFL network motif showing that this BN is the only one matching with the network motif and able to reproduce the behavior observed with quantitative models and synthetically-designed circuits. Note that in network suggested by the reviewer, node 2 is transformed into an inhibitor of 3 (instead of activator), making it a coherent FFL.

We included a new paragraph (p4, lines 265-272) explaining this result.

Comment 3

On pg2 col1 the authors write, "Therefore, the validity of a model cannot be assessed by the usual interpretations of BNs." The meaning of "usual interpretations" here (and in the following paragraph) should be clarified.

By "usual" we meant by using the synchronous/asynchronous interpretations, and all their sub-cases. We rephrased with mentioning explicitly the (a)synchronous term.

Comment 4

On pg3 col2 the authors write, "with an erroneous conclusion that its underlying influence graph is wrong" when discussing the failure of a BN to capture observed dynamic behavior (Fig. 2). This does not seem correct to me; a BN is not fully defined by its influence graph (Fig. 2a) but rather by its update rules (Fig. 2b). Similar to my comment 2, it seems to me here that one would first search for a different parameterization of the influence graph that succeeds in capturing the observed behavior.

Indeed, we made an unnecessary shortcut. We now replaced "underlying influence graph" by "logic" to avoid confusion.

Note that all logics compatible with the influence graph may fail with (a)synchronous analysis, whereas one succeeds with MP. And it is the case with the I3-FFL network motif, where an

(a)synchronous analysis would mis-conclude that the network is not sufficient. As already mentioned above, we included a new paragraph (p4, lines 265-272) explaining this result.

Comment 5

On pg4 col1, the authors write, "A component i can change to the increasing (resp. decreasing)..." and then later "Once in increasing (resp. decreasing) state, it can reach 1 (resp. 0) at any time." This second quote makes it clear that the first quote is referring to changing from state 0 to increasing and changing from state 1 to decreasing. I suggest changing the first quote to make this explicit.

Indeed, it might be useful to indicate near this text what transitions are possible between the 4 states (0, inc, dec, 1), as the text does not address whether it is possible to go back from increasing to 0 and from decreasing back to 1. (A table might work well.)

Moving to the "derivative of a continuous model" framework, one could argue that you start by going from 1 (a maximum) to decreasing and proceed by moving back to 1 without first passing through a minimum (0) and then a region with a positive slope (increasing). However, in the Boolean reduction one could consider such fluctuations occurring uniformly above or below the activation/deactivation threshold. Thus some interpretation here would be useful.

Either 0 or decreasing states can change to increasing, and either 1 or increase states can change to decreasing. It is however impossible to go directly from increasing to 0 without going through the decreasing state, and from decreasing to 1 without going through the increasing state.

We added this precision to the paragraph (p4, lines 243-254). Fig 3 recapitulates the possible transitions between the 4 states.

Therefore, a fluctuating system can be simulated in MPBNs both by a steady state at 1 (i.e., the fluctuation stay above all the interactions thresholds) and by oscillations $1 \rightarrow \text{decreasing} \rightarrow \text{increasing} \rightarrow 1$ without having to pass through the 0 state, but still matching the slopes.

Comment 6

On pg5 col1 the authors introduce some notation (e.g. $P^{\{NP\}}$), which should be defined for readers who are unfamiliar with it.

This complexity class denotes problems that can be solved in polynomial time, assuming problems of class NP can be solved in one instruction (known as an oracle). $P^{\{NP\}}$ is known to be harder than NP, but simpler than PSPACE, i.e., simpler than reachability and attractors with (a)synchronous BNs.

We modified the relevant paragraph to improve the explanation of this complexity class (p5, lines 352-357).

Comment 7

In the SI it appears that several Latex references are broken (e.g. the beginning of section 3).

We apologize for this mistake of compilation of the LaTeX document.

Reconciling Qualitative, Abstract, and Scalable Modeling of Biological Networks

Loïc Paulevé^{a,b,✉}, Juraj Kolčák^c, Thomas Chatain^c, and Stefan Haar^c

^aUniversité Bordeaux, Bordeaux INP, CNRS, LaBRI, UMR5800, Talence, France

^bLRI UMR8623, Université Paris-Sud, CNRS, Université Paris-Saclay, France

^cInria and LSV, CNRS (UMR 8643) and ENS Paris-Saclay, Université Paris-Saclay, France

**Predicting biological systems’ behaviors requires taking into account many molecular and genetic elements for which limited information is**
**available past a global knowledge of their pairwise interactions. Logical modeling, notably with Boolean Networks (BNs), is a well-established**
**approach that enables reasoning on the qualitative dynamics of networks. Several dynamical interpretations of BNs have been proposed.**
**The synchronous and (fully) asynchronous ones are the most prominent, where the value of either all or only one component can change at**
**each step. Here we prove that, besides being costly to analyze, these usual interpretations can preclude the prediction of certain behaviors**
**observed in quantitative systems. We introduce a new paradigm, the Most Permissive Boolean Networks (MPBNs), which offer the formal**
**guarantee not to miss any behavior achievable by a quantitative model following the same logic. Moreover, MPBNs significantly reduce the**
**complexity of dynamical analysis, enabling to model genome-scale networks.**

computational modeling | automatic reasoning | simulations | systems biology

Correspondence (✉): loic.pauleve@labri.fr

**M**odels in systems biology typically integrate knowledge
and hypotheses on molecular interactions, manually or
semi-automatically, gathered from experimental data found
in databases and the literature. These models are often quali-
fied as “mechanistic,” in opposition to those solely based on
biophysical laws.
Since their introduction in the late ’60s (1, 2), logical mod-
els, such as Boolean Networks (BNs), have been widely adopted
for reasoning about signaling and gene networks (3–11) as they
require few parameters and can easily integrate information
from omics datasets and genetic screens. These models rep-
resent processes with a high degree of generalization and can
offer coarse-grained but robust predictions. That makes them
particularly suitable for large biological networks, for which
ample global knowledge exists about potential interactions
with little precise data on actual molecules abundances and
reaction kinetics.
The validation of computational models is necessary to
trust their subsequent predictions. In systems biology, vali-
dation primarily involves *in silico* reproduction of observed
behaviors by executing the computational model. Such obser-
vations may be measurements of the activity, over time, or
at steady-state, of some of the interacting molecules under
different experimental conditions. Therefore, if no executions
of a BN reproduce an experimentally observed behavior (e.g.,
the activation of a particular gene), the model, and the asso-
ciated interactions, is considered as invalid. This procedure
also enables general studies on interaction motifs that are
necessary or sufficient for achieving fundamental behaviors
such as cellular differentiation or homeostasis (12–15).
~~Boolean Networks are often created from scratch~~
~~and rather than derived from a detailed mechanistic~~
~~(partially-parameterized) model. Consequently, there is no~~
~~guarantee that their analysis can be relevant for a more~~
~~precise model, and thus for the actual biological system.~~
~~A BN specifies the logic of activation of each component (or~~

~~node) of the system and aims at abstracting away quantitative~~
~~aspects related to kinetics and molecule abundances. For~~
~~instance, it may specifies that component c can turn “on”~~
~~whenever its activator b is “on” provided its inhibitor a is “off”.~~
~~Considering that the activity of components in the underlying~~
~~system is not binary, the “on” and “off” actually relate to~~
~~activity/abundance of molecules being above or below an~~
~~interaction threshold. However, one may wonder whether~~
~~such a binary coarse-graining may impede the validation of~~
~~the model, leading to reject a BN whereas it describes the~~
~~logic of components’ activities correctly.~~

Fig. 1 illustrates this issue with the incoherent feed-
forward loop of type 3, I3-FFL(16). ~~An (16), where an~~
~~input node 1 directly inhibits the output 3, but indirectly~~
~~activates it via node 2. Theoretical studies~~
~~The logic of nodes’~~
~~activation is fixed: the activation of 3 requires that node 2~~
~~is sufficiently active and that node 1 is not sufficiently active.~~
~~Theoretical studies with quantitative models (17, 18) and ex-~~
~~perimental data from synthetically designed circuits show~~
~~that (19) showed that, depending on kinetics parameters and~~
~~starting from all nodes being inactive, a monotonic activation~~
~~of the input can lead to a transient activity of the output~~
~~(node 3). However, it is impossible to reproduce this behavior~~
~~with usual (a)synchronous interpretations of BNs, including~~
~~synchronous and asynchronous: if 1 is not active, neither 2~~
~~nor 3 can be activated. If 1 is active, 2 is active, but any~~
~~transient activation of 3 is prevented (Fig. 1(d)).~~

Additional model features, such as intermediate levels for
the nodes, or delays in interactions, would allow a transient
activation for the I3-FFL output. However, such features come
with additional parameters and higher computational cost,
which limits their general application to large scale networks.

This simple example seems to show that setting binary
activities for nodes can both generate spurious behaviors
(as expected with qualitative models), and also preclude
the ~~discovery-recapitulation~~ of existing behaviors. There-

Fig. 1. Incoherent feed-forward loop of type 3 (a) and its associated Boolean logic for nodes activities (b); $f_{1,2,3}(x)$ are the Boolean functions used to compute the next value of each node from a given configuration x of the network, which is here a binary vector specifying the current value of each node, x_i referring to the Boolean value of node i . Whereas theoretical and experimental studies show a possible showed that starting from all nodes being inactive, an activation of the output is possible when the signal is active turned on (c), usual BNs analysis cannot predict this transient behavior: (d) shows the corresponding complete dynamics of f -starting from the configuration where configurations all nodes are inactive, and signal is set to 1. Configurations are represented by piles of 3 squares, where the top square represents the state of the first component, and so forth. A white square represents the inactive (0) state; a blue square represents the active (1) state; a dashed line indicates that no further evolution is possible. Arrows indicate possible transitions. The node 3 is never predicted to be active.

fore, the validity of a model cannot be assessed by the usual (a)synchronous interpretations of BNs. This limitation largely impedes the inference of dynamical network models and the identification of necessary interaction network motifs since the Boolean interpretation can strongly distort the landscape of candidate models wrongly conclude that no BN matching with a network motif can reproduce the desired behavior.

However, we found that this issue is actually due to the usual interpretations of BNs and not to their intrinsic Boolean nature. We introduce a new simulation approach, the *Most Permissive Boolean Networks* (MPBNs), which presents the formal guarantee to capture all behaviors achievable without the need for additional parameters. If MPBNs cannot reproduce a given observation, no quantitative refinement of the Boolean model can do it, and the model can safely be considered as incoherent with the observations. While predicting more behaviors than the usual synchronous and asynchronous interpretations of Boolean Networks, MPBNs still capture essential dynamical features of biological models.

Moreover, we demonstrated that the analysis of MPBNs avoids the state space explosion problem, a strong limiting factor for the usual synchronous and asynchronous interpretations of BNs. The drastically reduced computational cost enables the precise qualitative analysis of dynamics of genome-scale networks.

Modeling with Boolean Networks
Computational modeling of dynamical systems relies on two
fundamental ingredients: a language to specify the model, and
an execution semantics. The language provides symbols and
syntax rules to write a model, while the execution semantics
mathematically defines how to interpret it. The semantics
formalizes the notion of network configurations (or states)
and how to compute their evolution over time. It provides

an exhaustive assessment of model capabilities by enabling
dynamical analyzes such as simulations as well as formal veri-
fication by invariant analysis and model-checking. 108

A BN is specified by a mathematical function mapping any
binary vector of dimension n to another binary vector of the
same dimension: 111

$$f : \mathbb{B}^n \rightarrow \mathbb{B}^n \quad [1] \quad 112$$

where $\mathbb{B} = \{0, 1\}$ represent the Boolean values. Each element
of a binary vector models the state (inactive/active, absent/p-
resent) of the associated network node, and f_i is the function
which specifies the state towards which the i -th element evolves.
Fig. 2(b) gives an example of a BN modeling a switch system. 117

BNs semantics computes the possible temporal evolutions
of the component states using different methods. With *syn-*
*chronous* executions of BNs (introduced by S. Kauffman (1)),
we update all the components of the network at the same
time, and a configuration $x \in \mathbb{B}^n$ can only evolve to one con-
figuration $f(x)$. With *fully asynchronous* executions of BNs
(introduced by R. Thomas and usually referred to more simply
as *asynchronous* in the computational systems biology litera-
ture), we update only one component at a given time, and a
configuration $x \in \mathbb{B}^n$ can evolve to any configuration which
differs only by a single component i where $f_i(x) \neq x_i$. This
introduces potential *non-determinism* in the model trajectory
since there can be different executions of the same BN from a
given initial configuration. The (fully) asynchronous seman-
tics is often described as more realistic for modeling biological
networks, accounting for different kinetics of interactions. 133

Many more variants of executions of BNs have been studied
in the literature, some imposing a precise order in the updating
of the components, others allowing subsets of components to
be updated simultaneously, etc. Most, if not all, generate a
subset of the executions achievable with the (generalized) *asy-*
*nchronous* semantics of BNs where any number of components
can be updated at a time: a configuration can evolve to any
other configuration that complies with the logical functions
for the components that differ between both. Formally, for
any $x, y \in \mathbb{B}^n$, 143

$$x \xrightarrow{f_a} y \iff \forall i \in \Delta(x, y), y_i = f_i(x) \quad [2] \quad 144$$

where $\Delta(x, y)$ is the list of components which state differs
145 between x and y , i.e., $\Delta(x, y) = \{i \in \{1, \dots, n\} \mid x_i \neq y_i\}$. 146

A configuration $y \in \mathbb{B}^n$ is *reachable* from $x \in \mathbb{B}^n$ if either
147 $x = y$, or there exists a sequence of transitions from x to y : 148

$$\rho_a^f(x) = \{y \in \mathbb{B}^n \mid x = y \text{ or } x \xrightarrow{f_a} \dots \xrightarrow{f_a} y\} \quad [3] \quad 149$$

Notice that if $y \notin \rho_a^f(x)$, then it is impossible to evolve from x
150 to y according to any of the semantics defined above, including
151 the synchronous and (fully) asynchronous ones. 152

[revised manuscript text omitted]

Fig. 3 ~~illustrates~~ summarizes the changes of component states possible with the Most Permissive semantics. A component i can change to the ~~increasing~~ (resp. ~~decreasing~~) state \nearrow (resp. \searrow) state from the 0 or \searrow (resp. 1 or \nearrow) state whenever it can interpret the value of its regulators in a way which makes its logical function f_i true (resp. false) – if one of its regulators is in a dynamic state, both Boolean interpretations can be considered. Once in ~~increasing~~ (resp. ~~decreasing~~) state, it can reach 1 (resp. 0) at any time. ~~As a result, a component cannot go from~~ (resp. \searrow) state to 0 (resp. 1) without going through the \searrow (resp. \nearrow) state. Each component evolves independently of all others. The complete formal definition is given in SI.2.

Fig. 4 shows an example of execution using the Most Permissive semantics on the BN of Fig. 2. Contrary to the (a)synchronous interpretations, the Most Permissive semantics correctly captures the possible (transient) reachability of the configuration where the three genes are active. While component 1 is “increasing” and component 2 is active, gene 3 can indeed change to “increasing”, thus leading to the activation of all three components. This configuration is not in an attractor, and both single-point attractors identified in Fig. 2(c) are reachable via different Most Permissive executions. ~~We~~

~~We provide in Fig. S1~~ the application of MPBNs to the BN

Fig. 4. One of the possible executions using the Most Permissive semantics on the Boolean network in Fig. 2(b) starting from the configuration where all genes are inactive. Note that it correctly recovers the (transient) reachability of the configuration where the three components are active.

~~of the I3-FFL motif presented in Fig. 1 in Fig. S1., which successfully captures the transient activation of node 3. Even if we allow changing the Boolean logic, it is the only BN that can reproduce the observed transient and steady behaviors (SI.3.D). Therefore, a Boolean asynchronous analysis would have concluded that the network motif is insufficient to reproduce the observed behavior.~~

Formal guarantees for model refinements. Using the simple examples in Fig. 1 and Fig. 2, we have shown that BN refinements can introduce behaviors that cannot be captured with classical semantics.

Most Permissive Boolean Networks bring the formal guarantee of being able to reproduce *all* the behaviors achievable in *any* refinements, being a multivalued network or an ODE system (Theorem 1 and Corollary 1 in SI.2). In other words, if the Most Permissive semantics concludes that it is impossible to observe a given state change for some components, then no qualitative or quantitative model verifying the refinement criteria can predict these state changes.

The refinement criterion relies on a *binarization* of the multivalued configuration. An appropriate binarization necessarily quantifies 0 as Boolean 0 and m as 1, and is free for the other intermediate values. Let us denote by $\beta(x)$ the set of possible binarization of configuration $x \in \mathbb{M}^n$:

$$\beta(x) = \{x' \in \mathbb{B}^n \mid \forall i \in \{1, \dots, n\}, x_i = 0 \Rightarrow x'_i = 0 \text{ and } x_i = m \Rightarrow x'_i = 1\} . \quad [5]$$

For example with $m = 2$, $\beta(012) = \{001, 011\}$.

Then, we say a MN F is a refinement of a BN f of the same dimension n if and only if for every configuration $x \in \mathbb{M}^n$, and for every component $i \in \{1, \dots, n\}$, $F_i(x) < 0$ there exists $x' \in \beta(x)$ such that $f_i(x') = 0$, and $F_i(x) > 0$ implies there exists $x' \in \beta(x)$ such that $f_i(x') = 1$.

This characterization of BN refinement to MN can be directly extended to ODEs. Indeed, ODEs specify the (real) derivative of the (positive real) value of each component:

$$\mathcal{F} : \mathbb{R}_{\geq 0}^n \rightarrow \mathbb{R}^n . \quad [6]$$

Only the binarization β should be adapted in Eq. (5) to reflect that there is no (a priori) upper bounded value m for components.

The completeness property states the following. Consider a multivalued refinement F of a BN f with which there exists an asynchronous trajectory from a multivalued configuration x to y . Let us write \hat{x} any most-permissive configuration *compatible* with x : if $x_i = 0$, then $\hat{x}_i = 0$, if x_i is the maximum value of i , then $\hat{x}_i = 1$, and in the other cases \hat{x}_i can be either \nearrow or \searrow . Then, there exists a most-permissive trajectory leading to any of these \hat{x} to a most-permissive configuration \hat{y} compatible

321 with y and which is consistent with the the changes between
322 x and y : $\hat{y}_i = \nearrow$ if $y_i > x_i$ and $y_i < m$, $\hat{y}_i = \searrow$ if $y_i < x_i$ and

[revised manuscript text omitted]
 types (Fig. S3). After booleanization (31), MPBNs efficiently handled the original large multivalued model of 100 species, whereas subtypes. Due to the large size of the model (101 components), the original study had to perform approximations through model reduction. Therefore, the Most Permissive interpretation of BNs is still stringent enough to capture processes that control reachable attractors and symbolic model-checking techniques, avoiding the need for computing attractors. On the other hand, MPBNs can efficiently handle the booleanized (31) original large multivalued model, list the attractors and compute their reachability following the input condition changes. The attractor computation enables determining that in most conditions the attractors are fixed points (and thus are identical in asynchronous BNs), in two conditions (APC and proTh1), the MPBN has one complex attractor, indicating the existence of at least one complex asynchronous attractor. Then, the MP reachability analysis concludes on the same reprogramming graph, at much lower computational cost.

In conclusion, as stated in previous sections, MPBNs are formally guaranteed never to ignore behaviors hidden by artifacts of usual Boolean modeling while still being specific enough to predict differentiation processes to capture behaviors that only multi-level discrete models could capture with (a)synchronous interpretations; and as supported by these case studies, the MP interpretation of BNs is still stringent enough to capture processes that control reachable attractors, and doing so at a much lower computational cost.

Discussion

The choice of the dynamical interpretation of BNs has drastic effects on their predictions. Whereas the (fully) asynchronous BN interpretation is often advised for practical applications, it overlooks behaviors emerging from different timescales for the interactions, leading to biases when selecting plausible models. Such misses are due to artifacts of configurations updates. On the contrary, MPBNs offer a framework for reasoning on the qualitative dynamics without making any strong *a priori* hypothesis about the timescale and thresholds of interactions, and without additional parameter.

The state-space explosion triggered by the usual synchronous and asynchronous interpretations of BNs is another significant bottleneck for their application in systems biology (3, 32). MPBNs offer drastic gains in computational complexity when analyzing possible trajectories and attractors, both elementary and essential properties, underpinning the potential of a model. In practice, the verification of these properties with asynchronous BNs is typically limited to networks with 50 to 100 nodes. On the contrary, deciding the reachability and attractor properties in MPBNs relies on scalable algorithms and does not suffer from the state-space explosion. For the case of locally-monotonic BNs, which is a classical hypothesis for biological networks, the complexity allows addressing very large scale networks, as illustrated in SI.3, with experiments on BNs with up to 100,000 components. Our software tool mpbn is available at and integrated in the CoLoMoTo notebook environment (33).

The prediction of attractors reachable from specific initial
conditions, and possibly under various mutant conditions, is at
the core of many studies using logical models. While MPBNs
can identify the complete set of reachable attractors several
orders of magnitude faster than asynchronous BNs, the quan-
tification of the propensities of each attractor, e.g., performed
by sampling the trajectories (23, 30), is yet to be explored. In
addition to the validation of and the control of predictions from
genome-scale models, the complexity breakthrough brought
by MPBNs together with their ability to overcome artifacts
of Boolean modeling paves the way towards the inference and
learning of large-scale logical models from experimental data.
**Software and data availability.** Our software tool `mpbn`
implementing reachability and attractor analysis in
BNs with Most Permissive interpretation is available at
github.com/pauleve/mpbn and [doi:10.5281/zenodo.3715516](https://doi.org/10.5281/zenodo.3715516),
and is integrated in the `CoLoMoTo` notebook environment
(33) available at colomoto.org/notebook. Notebooks
for reproducing the case studies are available at
[doi:10.5281/zenodo.3936123](https://doi.org/10.5281/zenodo.3936123), with instructions for their
execution.
**Acknowledgments.** This research was funded by the French
Agence Nationale pour la Recherche (ANR) in the context of ANR-
FNR project “AlgoReCell” (ANR-16-CE12-0034)
**Authors’ contributions.** LP, TC, SH designed the research; LP
defined MPBNs and demonstrated Theorems 1,3,4, implemented
code, performed experiments, and initially drafted the manuscript;
JK demonstrated Theorem 2; LP, JK, TC, SH wrote the manuscript.
The authors declare having no conflict of interest.
References
- 1. S. A. Kauffman. Metabolic stability and epigenesis in randomly constructed genetic nets. *Journal of Theoretical Biology*, 22:437–467, 1969. [doi:10.1016/0022-5193\(69\)90015-0](https://doi.org/10.1016/0022-5193(69)90015-0).
- 2. R. Thomas. Boolean formalization of genetic control circuits. *Journal of Theoretical Biology*, 42(3):563–585, 1973. [doi:10.1016/0022-5193\(73\)90247-6](https://doi.org/10.1016/0022-5193(73)90247-6).
- 3. N. Le Novère. Quantitative and logic modelling of molecular and gene networks. *Nature reviews. Genetics*, 16:146–158, 2015. [doi:10.1038/nrg3885](https://doi.org/10.1038/nrg3885).
- 4. S. Kauffman, C. Peterson, B. Samuelsson, and C. Troein. Random Boolean network models and the yeast transcriptional network. *Proceedings of the National Academy of Sciences*, 100(25):14796–14799, 2003. [doi:10.1073/pnas.2036429100](https://doi.org/10.1073/pnas.2036429100).
- 5. R. Zhang, M. V. Shah, J. Yang, S. B. Nyland, X. Liu, J. K. Yun, R. Albert, and T. P. Loughran. Network model of survival signaling in large granular lymphocyte leukemia. *Proceedings of the National Academy of Sciences*, 105(42):16308–16313, 2008. [doi:10.1073/pnas.0806447105](https://doi.org/10.1073/pnas.0806447105).
- 6. Z. Mai and H. Liu. Boolean network-based analysis of the apoptosis network: Irreversible apoptosis and stable surviving. *Journal of Theoretical Biology*, 259(4):760–769, 2009. [doi:https://doi.org/10.1016/j.jtbi.2009.04.024](https://doi.org/10.1016/j.jtbi.2009.04.024).
- 7. P. Martínez-Sosa and L. Mendoza. The regulatory network that controls the differentiation of t lymphocytes. *Biosystems*, 113(2):96–103, 2013. [doi:https://doi.org/10.1016/j.biosystems.2013.05.007](https://doi.org/10.1016/j.biosystems.2013.05.007).
- 8. D. P. A. Cohen, L. Martignetti, S. Robine, E. Barillot, A. Zinovyev, and L. Calzone. Mathematical modelling of molecular pathways enabling tumour cell invasion and migration. *PLoS Comput Biol*, 11(11):e1004571, 2015. [doi:10.1371/journal.pcbi.1004571](https://doi.org/10.1371/journal.pcbi.1004571).
- 9. W. Abou-Jaoudé, P. T. Monteiro, A. Naldi, M. Grandclaudon, V. Soumelis, C. Chauviya, and D. Thieffry. Model checking to assess T-helper cell plasticity. *Frontiers in Bioengineering and Biotechnology*, 2, 2015. [doi:10.3389/fbioe.2014.00086](https://doi.org/10.3389/fbioe.2014.00086).
- 10. P. Traynard, A. Fauré, F. Fages, and D. Thieffry. Logical model specification aided by model-checking techniques: application to the mammalian cell cycle regulation. *Bioinformatics*, 32(17):1772–1780, 2016. [doi:10.1093/bioinformatics/btw457](https://doi.org/10.1093/bioinformatics/btw457).
- 11. S. Collombet, C. van Oevelen, J. L. Sardina Ortega, W. Abou-Jaoudé, B. Di Stefano, M. Thomas-Chollier, T. Graf, and D. Thieffry. Logical modeling of lymphoid and myeloid cell specification and transdifferentiation. *Proceedings of the National Academy of Sciences*, 114(23):5792–5799, 2017. [doi:10.1073/pnas.1610622114](https://doi.org/10.1073/pnas.1610622114).
- 12. R. Thomas and M. Kaufman. Multistationarity, the basis of cell differentiation and memory. i. structural conditions of multistationarity and other nontrivial behavior. *Chaos: An Interdisciplinary Journal of Nonlinear Science*, 11(1):170, 2001. [doi:10.1063/1.1350439](https://doi.org/10.1063/1.1350439).
- 13. C. Soulé. Graphic requirements for multistationarity. *Complexus*, 1(3):123–133, 2003. [doi:10.1159/000076100](https://doi.org/10.1159/000076100).
- 14. L. Paulevé and A. Richard. Static analysis of Boolean networks based on interaction graphs: a survey. *Electronic Notes in Theoretical Computer Science*, 284:93–104, 2011. [doi:10.1016/j.entcs.2012.05.017](https://doi.org/10.1016/j.entcs.2012.05.017).
- 15. S. Soliman. A stronger necessary condition for the multistationarity of chemical reaction networks. *Bulletin of Mathematical Biology*, 75(11):2289–2303, 2013. [doi:10.1007/s11538-013-9893-7](https://doi.org/10.1007/s11538-013-9893-7).
- 16. S. Mangan and U. Alon. Structure and function of the feed-forward loop network motif. *Proceedings of the National Academy of Sciences*, 100(21):11980–11985, 2003. [doi:10.1073/pnas.2133841100](https://doi.org/10.1073/pnas.2133841100).
- 17. S. Ishihara, K. Fujimoto, and T. Shibata. Cross talking of network motifs in gene regulation that generates temporal pulses and spatial stripes. *Genes to Cells*, 10(11):1025–1038, 2005. [doi:10.1111/j.1365-2443.2005.00897.x](https://doi.org/10.1111/j.1365-2443.2005.00897.x).
- 18. G. Rodrigo and S. F. Elena. Structural discrimination of robustness in transcriptional feedforward loops for pattern formation. *PLoS ONE*, 6(2):e16904, 2011. [doi:10.1371/journal.pone.0016904](https://doi.org/10.1371/journal.pone.0016904).
- 19. Y. Schaerli, A. Munteanu, M. Gili, J. Cotterell, J. Sharpe, and M. Isalan. A unified design space of synthetic stripe-forming networks. *Nature Communications*, 5(1), 2014. [doi:10.1038/ncomms5905](https://doi.org/10.1038/ncomms5905).
- 20. M. K. Morris, J. Saez-Rodriguez, P. K. Sorger, and D. A. Lauffenburger. Logic-based models for the analysis of cell signaling networks. *Biochemistry*, 49(15):3216–3224, 2010. [doi:10.1021/bi902202q](https://doi.org/10.1021/bi902202q).
- 21. R. Thomas and R. d’Ari. *Biological Feedback*. CRC Press, Boca Raton , Florida, USA, 1990.
- 22. B. B. Aldridge, J. Saez-Rodriguez, J. L. Muhlich, P. K. Sorger, and D. A. Lauffenburger. Fuzzy logic analysis of kinase pathway crosstalk in TNF/EGF/insulin-induced signaling. *PLoS Computational Biology*, 5(4):e1000340, 2009. [doi:10.1371/journal.pcbi.1000340](https://doi.org/10.1371/journal.pcbi.1000340).
- 23. G. Stoll, E. Viara, E. Barillot, and L. Calzone. Continuous time boolean modeling for biological signaling: application of Gillespie algorithm. *BMC Systems Biology*, 6(1):116, 2012. [doi:10.1186/1752-0509-6-116](https://doi.org/10.1186/1752-0509-6-116).
- 24. L. Glass and S. Kauffman. Logical analysis of continuous, non-linear biochemical control networks. *Journal of Theoretical Biology*, 39(1):103–129, 1973. [doi:10.1016/0022-5193\(73\)90208-7](https://doi.org/10.1016/0022-5193(73)90208-7).
- 25. H. de Jong. Modeling and simulation of genetic regulatory systems: A literature review. *Journal of Computational Biology*, 9:67–103, 2002. [doi:10.1089/10665270252833208](https://doi.org/10.1089/10665270252833208).
- 26. C. H. Papadimitriou. *Computational Complexity*. Addison-Wesley, 1995.
- 27. H. Klarner, A. Bockmayr, and H. Siebert. Computing maximal and minimal trap spaces of Boolean networks. *Natural Computing*, 14(4):535–544, 2015. [doi:10.1007/s11047-015-9520-7](https://doi.org/10.1007/s11047-015-9520-7).
- 28. R. Albert and A.-L. Barabási. Statistical mechanics of complex networks. *Reviews of Modern Physics*, 74(1):47–97, 2002. [doi:10.1103/revmodphys.74.47](https://doi.org/10.1103/revmodphys.74.47).
- 29. E. Remy, S. Rebouissou, C. Chauviya, A. Zinovyev, F. Radvanyi, and L. Calzone. A modeling approach to explain mutually exclusive and co-occurring genetic alterations in bladder tumorigenesis. *Cancer Research*, 75(19):4042–4052, aug 2015. [doi:10.1158/0008-5472.can-15-0602](https://doi.org/10.1158/0008-5472.can-15-0602).
- 30. N. D. Mendes, R. Henriques, E. Remy, J. Carneiro, P. T. Monteiro, and C. Chauviya. Estimating attractor reachability in asynchronous logical models. *Frontiers in Physiology*, 9, 2018. [doi:10.3389/fphys.2018.01161](https://doi.org/10.3389/fphys.2018.01161).
- 31. G. Didier, E. Remy, and C. Chauviya. Mapping multivalued onto Boolean dynamics. *Journal of Theoretical Biology*, 270(1):177–184, 2011. [doi:10.1016/j.jtbi.2010.09.017](https://doi.org/10.1016/j.jtbi.2010.09.017).
- 32. S. Bornholdt. Boolean network models of cellular regulation: prospects and limitations. *Journal of The Royal Society Interface*, 5(suppl_1), 2008. [doi:10.1098/rsif.2008.0132.focus](https://doi.org/10.1098/rsif.2008.0132.focus).
- 33. A. Naldi, C. Hernandez, N. Levy, G. Stoll, P. T. Monteiro, C. Chauviya, T. Helikar, A. Zinovyev, L. Calzone, S. Cohen-Boulakia, D. Thieffry, and L. Paulevé. The CoLoMoTo Interactive Notebook: Accessible and Reproducible Computational Analyses for Qualitative Biological Networks. *Frontiers in Physiology*, 9:680, 2018. [doi:10.3389/fphys.2018.00680](https://doi.org/10.3389/fphys.2018.00680).

Reviewers' Comments:

Reviewer #1:

Remarks to the Author:

The revised manuscript successfully addresses all my suggestions. I recommend its acceptance.

I have a single suggested word change. In line 41, "reject a BN whereas it describes", I would replace "whereas" with "although".

Reviewer #2:

Remarks to the Author:

Dear Editor and Authors,

In their revisions to NCOMMS-20-12959A, the authors have thoroughly responded to the comments raised in my initial review. I now recommend the manuscript be accepted for publication once the authors have had the opportunity consider the following (very minor) follow-up comments.

Comment 1

In the paragraph ending on line 63, perhaps change the text "...if 1 is not active, neither 2 nor 3 can be activated." to "...starting from the state where all nodes are inactive, neither 2 nor 3 can be activated without the prior activation of 1." ?

This may be redundant because the authors have added text above to specify that all nodes begin inactive, but to my eye reinforcing this point here will help to reinforce this important point.

Comment 2

On line 279 the authors refer to "SI.3.D" However, in reviewing the SI it appears that section 3 only includes sections A-C. It appears rather that the authors are referring to section 4 of the SI.

Comment 3

Typo, line 39: "it may specifies" -> "it may specify"

REVIEWERS' COMMENTS:

Reviewer #1 (Remarks to the Author):

The revised manuscript successfully addresses all my suggestions. I recommend its acceptance.

I have a single suggested word change. In line 41, "reject a BN whereas it describes", I would replace "whereas" with "although".

We adopted this suggestion.

Reviewer #2 (Remarks to the Author):

Dear Editor and Authors,

In their revisions to NCOMMS-20-12959A, the authors have thoroughly responded to the comments raised in my initial review. I now recommend the manuscript be accepted for publication once the authors have had the opportunity consider the following (very minor) follow-up comments.

Comment 1

In the paragraph ending on line 63, perhaps change the text "...if 1 is not active, neither 2 nor 3 can be activated." to "...starting from the state where all nodes are inactive, neither 2 nor 3 can be activated without the prior activation of 1." ?

This may be redundant because the authors have added text above to specify that all nodes begin inactive, but to my eye reinforcing this point here will help to reinforce this important point.

We adopted this suggestion.

Comment 2

On line 279 the authors refer to "SI.3.D" However, in reviewing the SI it appears that section 3 only includes sections A-C. It appears rather that the authors are referring to section 4 of the SI.

We corrected the SI so the section is now SI.3.D.

Comment 3

Typo, line 39: "it may specifies" -> "it may specify"

We corrected this typo.